# Screening and Validation of Rhizobial Strains for Improved Lentil Growth

**DOI:** 10.3390/microorganisms13061242

**Published:** 2025-05-28

**Authors:** Tianda Chang, Tao Yang, Meng Ren, Xinghui Li, Xuerui Fang, Bingjie Niu, Hongbin Yang, Lixiang Wang, Ximing Chen

**Affiliations:** 1Corn Research Institute, Shanxi Agricultural University, Xinzhou 034000, China; z20223218@stu.sxau.edu.cn; 2College of Agronomy, Shanxi Agricultural University, Taigu, Jinzhong 030801, China; yangtao40@sxau.edu.cn (T.Y.); renmeng@sxau.edu.cn (M.R.); lixinhui@sxau.edu.cn (X.L.); z20223106@stu.sxau.edu.cn (X.F.); yanghongbin@sxau.edu.cn (H.Y.); 3College of Resource and Environment, Shanxi Agricultural University, Taigu, Jinzhong 030801, China; niubingjie1@sxau.edu.cn; 4Shanxi Houji Laboratory, Taiyuan 030031, China

**Keywords:** lentil, isolation and characterization, rhizobium, *nifH*, inoculation

## Abstract

Lentil is a nutritionally valuable legume crop, rich in protein, carbohydrates, amino acids, and vitamins, and is also used as green manure. Symbiotic nitrogen fixation (SNF) plays a crucial role in lentil growth and development, yet there is limited research on isolating and identifying lentil rhizobia related to nodulation and nitrogen fixation. This study employed tissue block isolation, line purification, and molecular biology to isolate, purify, and identify rhizobial strains from lentils, analyzing their physiological characteristics, including bromothymol blue (BTB) acid and alkali production capacity, antibiotic resistance, salt tolerance, acid and alkali tolerance, growth temperature range, and drought tolerance simulated by PEG6000. Additionally, the nodulation capacity of these rhizobia was assessed through inoculation experiments using the identified candidate strains. The results showed that all isolated rhizobial strains were resistant to Congo red, and *nifH* gene amplification confirmed their potential as nitrogen fixers. Most strains were positive for H_2_O_2_ and BTB acid and base production, with a preference for alkaline environments. In terms of salt tolerance, the strains grew normally at 0.5–2% NaCl, and six strains were identified as salt stress resistant at 4% NaCl. The temperature range for growth was between 4 °C and 49 °C. Antibiotic assays revealed resistance to ampicillin and low concentrations of streptomycin, while kanamycin significantly inhibited growth. Two drought-tolerant strains, TG25 and TG55, were identified using PEG6000-simulated drought conditions. Inoculation with candidate rhizobial strains significantly increased lentil biomass, highlighting their potential for enhancing crop productivity.

## 1. Introduction

Lentil (*Lens culinaris* Medik) is one of the earliest crops domesticated in the Fertile Crescent and spread to other regions during the Bronze Age, making it an ideal model for studying rhizobia evolution in legume crops [1]. In China, lentil is a significant small bean crop, primarily cultivated in regions such as Ningxia, Shanxi, Shaanxi, and Gansu [2]. Lentils are known for their high content of protein, carbohydrates, iron, calcium, phosphorus, magnesium, and vitamins A and B, offering high nutritional value with low fat content [3]. Consequently, the development and research of lentils hold significant theoretical and practical importance. Beyond serving as a dietary supplement to staple foods, lentils can be used in soups, salads, and as fillings for various dishes, thereby enriching dietary options [4]. Therefore, further research and development of lentils can contribute to enhancing agricultural productivity and improving human health.

Nitrogen is one of the main nutrient elements required for plants and is essential for plant growth and development. However, the nitrogen content in the soil often fails to meet the growth needs of most crops. So, nitrogen fertilizer is usually required to improve the crop yield [5]. The application of nitrogen fertilizer can effectively promote the growth of crops, but its excessive application also brings serious environmental problems, including soil compaction, water eutrophication, and the enrichment of heavy metals, which pose potential threats to land health and food security. Therefore, finding economical and environmentally friendly nitrogen sources has become an urgent task in the current agricultural production [6,7]. Developing new nitrogen sources, optimizing fertilization strategies, and promoting biological nitrogen fixation technology can effectively reduce the dependence on chemical nitrogen fertilizer, reduce its impact on the environment, and promote the coordinated development of agricultural production and environmental protection.

Rhizobia establish a symbiotic relationship with legumes by colonizing plant roots, forming nodules that convert atmospheric nitrogen into nitrogen compounds available to the plant. This not only significantly enhances the nitrogen supply for legume crops but also reduces their dependence on fertilizers, thereby helping to decrease environmental pollution. Inoculation with rhizobia can significantly improve the legume crop growth rate, enhance stress resistance, and improve the crop yield and quality [8,9,10]. In addition, the inoculation of rhizobia can also improve the soil structure, promote the diversity of soil microorganisms, and further enhance the soil fertility and ecological function [11]. The nature of symbiotic nitrogen fixation makes legumes one of the important crop rotation and soil improvement plants. This efficient nitrogen fixation system produces about 60% [12,13] of the total nitrogen fixation capacity. Therefore, the application of rhizobia in agriculture, and for the sustainable development of agriculture, is very important.

The current situation of scientific research and production of lentils as an important legume crop in China can be summarized [14]. Compared with other edible beans in China, there are very few studies on lentils, and they are limited to germplasm resources and cultivation techniques, while there are few reports on the commensal rhizobia of lentil. This study aimed to isolate, purify, and identify lentil rhizobia in Jinzhong District of Shanxi Province, evaluate the physiological and biochemical characteristics of the isolated rhizobia, and preliminarily test the ability of nodulation by the retrieval test.

## 2. Materials and Methods

### 2.1. Soil Samples and Seed Source

Soil samples were collected from two locations in Taigu District, Jinzhong City, Shanxi Province: the lentil-grown experimental fields in the Agricultural Crops Station (37.42157° N, 112.57109° E) and in Shenfeng Village (37.43275° N, 112.60202° E) of Taigu District. Six points of soil samples in each field were collected at a 20 cm depth of surface soil. The lentil variety used in the experiment was the British medium green lentil, collected in the Corn Research Institute, Shanxi Agricultural University.

### 2.2. Culture Medium

Initially, YMA (Yeast Mannitol Agar) medium was prepared using the following components: 10 g mannitol, 3 g yeast extract, 0.1 g NaCl, 0.2 g MgSO_4_·7H_2_O, 0.25 g K_2_HPO_4_, 0.25 g KH_2_PO_4_, and 15 g agar powder, with distilled water added to a total volume of 1000 mL. The pH was adjusted to 6.8–7.0, these chemicals was sourced from Coolaber, Beijing, China. Next, Congo Red medium was prepared by adding 5 mL of a 0.5% Congo Red solution to 1000 mL of the YMA medium. Finally, BTB (Bromothymol Blue) medium was prepared by adding Bromothymol Blue to the YMA liquid medium to achieve a final concentration of 0.01%. The pH of the solution was then adjusted until it turned light yellowish-green. Drought experiments were conducted using polyethylene glycol 6000.

### 2.3. Test Method

#### 2.3.1. Isolation and Purification of Rhizobia

Mature lentil root nodules from two locations were surface-sterilized using a modified protocol proposed by Peter et al. [15] in order to isolate and purify the strains, and ‘large’ and ‘full’ mature nodules on lentil roots from different soil samples were used to isolate, purify, and culture on the same day. Nodules were sterilized using UV light to ensure that the whole process was sterile. Nodules underwent sequential sterilization in a laminar flow hood: initial 10 min washing in sterilized distilled water to remove soil particles, followed by 5 min immersions in 75% ethanol and 5% sodium hypochlorite (NaClO), with final rinsing in sterile ddH_2_O. Using autoclave-sterilized forceps and scalpels, nodules were bisected to expose internal tissue, which was streaked onto Congo Red Yeast Mannitol Agar (CR-YMA) plates. The forceps were placed on the outer flame of an alcohol lamp and sterilized for about 1 min. Afterwards, the plates were inverted and incubated at 28 °C for 2–3 days, during which time rhizobial colonies were identified by their characteristic lack of Congo red absorption (remaining unstained or pale pink). Selected colonies underwent three successive purification cycles on fresh CR-YMA medium to obtain axenic cultures. Pure isolates were suspended in 30% glycerol solution and cryopreserved at −80 °C for long-term storage.

#### 2.3.2. Phenotypic Identification of Rhizobia

After 24–48 h of incubation in a constant temperature incubator at 28 °C, the colonies were examined for color. Strains that remained unstained or showed minimal coloration were considered positive, while those that absorbed the dye were deemed negative. Positive isolates identified based on their unstained morphology were subjected to subculturing in order to obtain axenic cultures. These purified strains were subsequently maintained at 4 °C for short-term preservation.

#### 2.3.3. Gene Identification of *Rhizobium* spp.

Genomic DNA was extracted following the method described by Gao [16]. The universal primers 27F and 1492R were used to amplify the 16S rDNA region of rhizobial strains. The PCR reaction mixture (25 μL) contained 12.5 μL Premix Taq, 0.5 μL each of 27F and 1492R primers, 0.5 μL template DNA, and ddH_2_O to volume. PCR conditions were initial denaturation at 94 °C for 5 min, 30 cycles of denaturation at 94 °C for 45 s, annealing at 55 °C for 45 s, and extension at 72 °C for 2 min, followed by a final extension at 72 °C for 10 min. PCR products were sequenced by Zhengzhou Biotech. The resulting sequences were analyzed and compared against the NCBI database, and selected 16S rDNA sequences were used to construct a Neighbor-Joining phylogenetic tree using MEGA11 software(Version 11.0.13). To confirm the presence of nitrogen-fixing genes, PCR amplification of the *nifH* gene was performed using specific primers, with the same reaction mixture as for 16S rDNA. The *nifH* PCR conditions were initial denaturation at 94 °C for 5 min, 30 cycles of denaturation at 94 °C for 40 s, annealing at 55 °C for 30 s, and extension at 72 °C for 1 min, followed by a final extension at 72 °C for 10 min. The *nifH* PCR products were visualized by electrophoresis on a 1% agarose gel. Primer sequences for both 16S rDNA and *nifH* amplification are provided in Table 1.

#### 2.3.4. Physiological and Biochemical Determination

(1)BTB acid and alkali production reaction

The test strains were inoculated into Bromothymol Blue (BTB) medium, and a parallel set of cultures was prepared in BTB-free medium to serve as the control group. Three independent biological replicates were performed. Cultures were incubated at 28 °C for 3 days. Results were interpreted as follows: a yellow color change indicated acid production and was considered a positive reaction, while a blue color change signified alkali production and was deemed a negative reaction [19].

(2)Catalase reaction

The test strain was inoculated onto agar plates and incubated at 28 °C. After 3 days of growth, 1 mL of 3% hydrogen peroxide (H_2_O_2_) solution was added directly to the bacterial colonies. The reaction was observed for 5 min. A positive result was identified by the rapid formation of gas bubbles on the surface of the colonies, indicating the presence of catalase activity. Conversely, if no visible gas bubbles were produced, the result was recorded as negative, meaning an absence of catalase in the tested strain. This experiment was conducted with three independent biological replicates to ensure reliability and reproducibility [20].

#### 2.3.5. Measurement of Antibiotic Tolerance

Antibiotic tolerance tests were conducted using nine different conditions. Three antibiotics were evaluated: kanamycin, ampicillin, and streptomycin. For each antibiotic, three concentrations were tested: 50 µg/mL, 100 µg/mL, and 200 µg/mL. The presence or absence of microbial growth on the solid medium was used as the primary indicator of bacterial survival under antibiotic conditions. The experiment was repeated three times to ensure reproducibility and statistical validity.

#### 2.3.6. Measurement of Stress Resistance

To assess the strain’s tolerance to various environmental conditions, a series of experiments were conducted using YMA medium as the base culture. Salt tolerance was evaluated by adjusting NaCl concentrations to 0.50%, 1%, 1.50%, 2%, and 4%. The strain’s ability to withstand acidic and alkaline conditions was tested by setting pH values at 4.0, 5.0, 7.0, 9.0, and 10.0. Additionally, the growth temperature range was examined by incubating cultures at 4 °C, 28 °C, 37 °C, and 49 °C, the presence of colonies in the medium was used as the index to show the stress tolerance, the primary criterion for determining tolerance under each condition was the presence or absence of a visible colony in the solid medium with stress treatments. All experiments were replicated three times.

#### 2.3.7. PEG6000 Tolerance Determination

The osmotic stress tolerance of the strains was evaluated using different concentrations of PEG6000 (polyethylene glycol 6000). The concentrations were set at 0%, 5%, 10%, 20%, and 30%, denoted as CK, PEG 5%, PEG 10%, PEG 20%, and PEG 30%, respectively [21]. The osmotic potential of these PEG6000 solutions was calculated using the formula developed by Michel et al. [22], resulting in values of 0, −0.06, −0.17, −0.53, and −1.10 MPa, respectively.

To assess growth under osmotic stress, all strains were initially cultured in sterilized YMA liquid medium at 28 °C with shaking at 200 rpm for 4 days to prepare the inoculum. Subsequently, 10 μL of this bacterial suspension was transferred into YMA liquid medium containing different concentrations of PEG6000. The cultures were then incubated at 28 °C with shaking at 200 rpm for 7 days. Growth was quantified by measuring the optical density (OD) at 600 nm, with higher OD values indicating better growth under the given conditions [23]. Data are presented as the average of three trials. This method allows quantitative analysis of the effect of different concentrations of PEG6000 on the growth of the test strain.

#### 2.3.8. Re-Inoculation Assay of Surface Soil Strains

Tolerant strains were selected based on their physiological and biochemical performance and identified through back-entry analysis. The experiment began with seed sterilization. Lentil seeds were surface-sterilized using 75% ethanol and 5% NaClO solution for 5 min, followed by 3 to 5 rinses with sterilized distilled water. The sterilized seeds were then placed on filter paper in Petri dishes and placed in a constant temperature incubator at 28 °C for incubation. Before lateral root development, lentil seedlings with uniform hypocotyl length were selected, and vermiculite and perlite were mixed and sterilized at a volume ratio of 2:1 and then packed into 10 cm diameter plastic cups. Two treatment groups were established: a control group without bacteria and a treatment group inoculated with the bacterial strains. When the lentil plants reached the developmental stage of fully expanded compound leaves, each plant was sprayed with either 30 mL of sterile distilled water (control treatment) or a freshly prepared bacterial suspension with OD600 = 0.08. Throughout the experimental period, all plants were irrigated regularly with a nitrogen-free nutrient solution designed to support plant growth without interfering with potential symbiotic nitrogen fixation. The composition of the nutrient solution (per liter) was as follows: 0.03 g/L Ca(NO_3_)_2_·4H_2_O, 0.1 g/L CaCl_2_·2H_2_O, 0.1 g/L KH_2_PO_4_, 0.15 g/L Na_2_HPO_4_·12H_2_O, 0.12 g/L MgSO_4_·7H_2_O, 0.05 g/L Ferric Citrate, 2.86 mg/L H_2_BO_3_, 1.81 mg/L MnSO_4_, 0.22 mg/L ZnSO_4_, 0.8 mg/L CuSO_4_, and 0.02 mg/L Na_2_MoO_4_·2H_2_O. After 30 days, nodulation was observed, and various plant growth parameters were measured, including aboveground fresh weight, underground fresh weight, root length, plant height, and root number for each plant. To ensure statistical reliability and reproducibility, each treatment group consisted of three independent biological replicates, with all measurements performed in a randomized block design.

### 2.4. Data Processing

Statistical evaluation of the dataset was conducted using IBM SPSS Statistics (Version 23) to ensure accuracy and consistency in data processing. For graphical representation and comparative analysis between groups, GraphPad Prism (Version 8.0.0.3) was employed to perform independent samples t-tests. The experimental design was based on a completely randomized approach and independently replicated across three biological experiments.

## 3. Results

### 3.1. Isolation and Purification of Rhizobium *spp.*

Sixty-five Rhizobium strains were successfully isolated from lentil plants grown in soil samples from the Agricultural Crops Station and in Shenfeng Village of Taigu District, the morphological analysis revealed that these strains exhibited colonies that were round or oval in shape, with a moist, opaque, and milky appearance, and were slightly convex (Figure 1). These characteristics align with the typical growth phenotypes of rhizobia, which are known for their round or oval colonies and mucoid texture. Based on these findings, a preliminary judgment suggests that the isolated strains are likely rhizobia, given their consistency with the basic characteristics of this genus.

### 3.2. Molecular Identification and Sequence Analysis of Rhizobia

The amplification of the *nifH* and 16S rDNA genes in the isolated strains resulted in a single clear band consistent with the expected size for most rhizobia (Figure 2). This process provided the basis for a more precise subsequent classification of the strains. Following sequencing, the obtained sequences were aligned against the NCBI online database, revealing high sequence identity with *Rhizobium* sp. and *Rhizobium leguminosarum.* The 16S rDNA sequences were used to construct a phylogenetic tree using the adjacency method in MEGA11 software. Notably, the results suggest that the bacteria isolated in this study may belong to the genus *Rhizobium* sp. and *Rhizobium leguminosarum* (Figure 3).

### 3.3. Measurement of Physiological and Biochemical Reactions

The physiological and biochemical responses of rhizobia are relevant for their classification and identification. Therefore, the physiological and biochemical responses of all strains were determined. Among all the strains listed in Table 2, only eight strains (TG 3, TG 6, TG 8, TG 10, TG 32, TG 60, TG 66, and TG 67) yielded a negative reaction with no air bubbles produced. In contrast, the remaining 57 strains produced positive reactions, with 57 strains exhibiting yellow coloration, and eight strains (TG 13, TG 16, TG 34, TG 39, TG 41, TG 44, TG 56, and TG 58) producing an alkali blue coloration. The physiological and biochemical tests revealed notable differences in the responses of the rhizobia strains, highlighting their diverse physiological and biochemical characteristics.

### 3.4. Antibiotic Resistance

Rhizobium resistance contributes to strain isolation, purification, and subsequent genetic manipulation. Antibiotic resistance assay of all strains in this study revealed that all strains demonstrated normal growth at ampicillin concentrations of 50 μg/mL and 100 μg/mL. However, when the concentration was increased, some strains exhibited intolerance at 200 μg/mL. Specifically, only two strains, TG 2 and TG 5, were intolerant to streptomycin at a concentration of 50 μg/mL. Most strains were unable to grow normally at streptomycin concentrations of 100 μg/mL and 200 μg/mL. Regarding kanamycin, some strains showed tolerance at 50 μg/mL, including TG 1, TG 3, TG 6, TG 7, TG 8, TG 13, and TG 67. When the kanamycin concentration was increased to 100 μg/mL and 200 μg/mL, only six strains (TG 3, TG 6, TG 7, TG 8, TG 13, and TG 67) were able to grow (Table 3).

### 3.5. Determination of Salinity Tolerance and Temperature Adaptability

Soil salinity has a significant impact on agricultural production and ecology, while temperature is also a key factor affecting the growth of rhizobia. Therefore, it is crucial to screen rhizobia with a wide range of salinity tolerance and temperature adaptation. According to Table 4, all strains were able to grow normally at a NaCl concentration of 0.5%, and the same strains also grew well at concentrations of 1% to 2% NaCl, with 16 strains showing tolerance at these levels. However, only six strains (TG 6, TG 7, TG 8, TG 13, TG 35, and TG 57) were able to grow at a higher concentration of 4% NaCl, indicating that these strains possessed high salt tolerance. In terms of pH tolerance, all strains grew normally in a neutral environment at pH 7. However, none of the strains could grow in an acidic environment at pH 4. Thirty-six strains were able to grow at pH 5, while 27 strains grew at pH 9, and 13 strains grew normally at pH 10. This suggests that the lentil rhizobia can adapt to alkaline environments. Regarding temperature adaptation, all strains were unable to grow normally at temperatures below 4 °C or above 49 °C. This indicates that the test strains were not adapted to either low-temperature or high-temperature environments.

### 3.6. PEG6000 Tolerance Determination

Drought and water scarcity significantly impact the establishment of nodulation and nitrogen fixation symbiosis between rhizobia and legumes. This study observed (Table 5) that the OD values of different strains in PEG6000 solution decreased with increasing concentration of PEG6000; that as the concentration of PEG6000 increased, the optical density (OD) values decreased by an average of 22.48%, 42.07%, 59.70%, and 86.86% at concentrations corresponding to 5%, 10%, 20%, and higher levels of drought stress, respectively. This suggests that the inhibition of strain growth increases with increasing drought. Notably, when the PEG6000 concentration reached 20%, 27 strains still maintained OD values above 60%, indicating severe growth inhibition at this level of drought. The comparison of OD value decreases among different strains revealed considerable variability under the same PEG6000 concentration. Therefore, according to the findings, after screening with strains TG 25 and TG 55, they showed significantly lower OD values compared to the others. These findings suggest that TG 25 and TG 55 could serve as useful experimental materials for improving drought tolerance in rhizobia.

### 3.7. The Reidentification of Candidate Rhizobia Can Significantly Improve the Nodulation and Growth Status of Lentils

Most strains tested positive for the bromothymol blue (BTB) and catalase tests, while TG 14 showed resistance to antibiotic concentrations. Strains TG 8, TG 13, and TG 17 were randomly selected for their strong resistance to acid–base conditions, salt, and temperature. Additionally, TG 55 was identified as the most drought-resistant strain. These five lentil rhizobia strains (TG 8, TG 13, TG 14, TG 17, and TG 55) were chosen as candidate strains for further testing. Inoculation experiments revealed that the numbers of root nodules, root length, fresh weight, and plant height were higher in inoculated plants compared to the control without inoculation. As shown in Figure 4, strains TG 8, TG 14, TG 17, and TG 55, when inoculated, increased various growth parameters of lentils. Specifically, TG 8, TG 14, and TG 17 enhanced root length, while TG 8, TG 17, and TG 55 significantly increased the ground fresh weight and height of lentils. Furthermore, TG 17 and TG 55 significantly improved the ground fresh weight of lentils when used as inoculants. Excluding strain TG 13, inoculation with rhizobia strains TG 8, TG 14, TG 17, and TG 55 improved nodulation or growth indices of lentils. Notably, strain TG 17 enhanced all growth indices of lentils, making it suitable for use in practical production as a basis for developing bacterial agent resources.

## 4. Discussion

In this study, 65 rhizobial strains were successfully isolated from the agricultural fields of Taigu District and Shenfeng Village, Jinzhong City, Shanxi Province. The colonies exhibited typical rhizobial morphology, appearing as round or oval, opaque, and slightly raised structures that did not stain with Congo red. These growth phenotypes align with the basic characteristics of rhizobia, consistent with previous findings [24,25]. As CR-YMA is not very specific, as it supports the growth of various microorganisms, which can lead to contamination and make it challenging to differentiate between similar-looking colonies without microscopic examination, other media suitable for rhizobium growth may promote the isolation of more lentil rhizobium. Molecular analysis through 16S rDNA sequencing and phylogenetic tree construction revealed that the isolates are closely related to *Rhizobium* sp. and *Rhizobium leguminosarum.* Notably, three of the four lineages of Cicer lentis were found to belong to *Rhizobium. leguminosarum*, supporting its role as the primary rhizobium associated with lentil [26]. Lentils are known to form a symbiotic relationship with *Rhizobium leguminosarum.* According to Peter Young [27], the rhizobia isolated from lentils may be attributed to *Rhizobium johnstonii*. This conclusion is supported by the fact that the type strain of *Rhizobium johnstonii* was originally isolated from *Pisum sativum.* Furthermore, lentils exhibit similarities to *Pisum sativum* regarding host specificity and are also capable of establishing a symbiotic relationship with *Rhizobium leguminosarum bv. vicia*. [28,29].

The screening of rhizobial strains with strong stress resistance is a prerequisite for establishing an excellent legume–biological nitrogen fixation symbiotic system. The physiological and biochemical determinations in this study were positive for catalase and BTB, which are generally consistent with the metabolic and physiological characteristics of rhizobia [30]. The test strains grew normally at low concentrations of ampicillin and streptomycin. However, when the concentration increased to 200 µg/mL, they exhibited intolerance, indicating that their antibiotic resistance decreased with increasing concentrations. Kanamycin significantly inhibited the growth of the strain [31]. In another study, only 4 out of 78 strains were found to be resistant to kanamycin at a concentration of 50 µg/mL [32]. The same strain displayed varying resistance levels to different antibiotics, and resistance to the same antibiotic weakened as the concentration increased [33,34]. In acid–base reactions, most strains tend to be more adapted to alkaline environments. A screening of lentil rhizobia in Bengal revealed that most strains thrive in alkaline media [35]. Similarly, Wei Ge Macro [36] isolated five types of Astragalus plant rhizobia from the Shanxi–Gansu region and found that these strains have a strong ability to resist alkalinity based on physiological and biochemical index determinations. In terms of salt concentration tolerance, most strains were able to grow at NaCl concentrations ranging from 0.5% to 2%, while only six strains grew normally at 4% NaCl [37]. It was observed that the strain grew normally at NaCl concentrations of 1–2% [38]. However, when the NaCl concentration increased to 4%, none of the strains could grow, indicating that high NaCl concentrations inhibited strain growth. In temperature determinations, the strains failed to grow normally at 4 °C and 49 °C, consistent with their lack of resistance to both low and high temperatures [39]. Nonetheless, Kang Wenjuan’s [40] study in three cultivated ecological areas of Gansu Province found that the strains were sensitive to low temperatures, suggesting that the living environment influences their stress resistance. In preliminary screenings for drought-resistant strains, Pei Xiaofeng et al. [41] identified seven drought-tolerant strains, verifying the feasibility of using PEG6000 for such screenings. This study isolated two drought-resistant strains, TG 25 and TG 55, through primary screening using a PEG6000 drought experiment. He Yahui [42] showed diversity in tolerance to drought stress, similar to the findings of this study. Strains that are tolerant to a wide range of stress factors are able to survive harsh environmental conditions, such as extreme temperatures, drought, and high salinity. They continue to perform their basic functions in the soil, such as decomposing organic matter and participating in the material cycle, preventing the collapse of soil ecosystem functions. By decomposing complex organic matter into simple inorganic matter, they enable the recycling of nutrients in the soil, providing other organisms with the nutrients they need to survive [43,44,45]. Moreover, they can be used in conjunction with organic fertilizers to decompose organic matter, release nutrients, provide comprehensive nutritional support for plants, and at the same time enhance the plant’s own resilience against pests and diseases, so as to guarantee the yield and quality of organic agricultural products [46,47].

The candidate strains were selected for further study and were found to form nodules with lentils effectively. Following inoculation with these rhizobia, significant increases were observed in plant height, aboveground fresh weight, belowground fresh weight, and the number of nodules compared to uninoculated controls. Wang Jinsheng et al. [48] isolated rhizobia strains 112-1 and 113-1 using YMA medium and found that soybeans inoculated with these rhizobia exhibited significantly greater plant height, single-strain number, and 100-grain weight. Similarly, Liu Ruirui et al. [49] isolated and purified the Hbu074005 rhizobia strain, which, when used in inoculation experiments, resulted in a significantly higher ear number in peanuts compared to other treatments. Although root length did not show significant improvement in this study, further trials are needed to explore this aspect. The candidate strains isolated in this study provide a solid foundation for developing lentil inoculant resources for practical agricultural applications.

## 5. Conclusions

A total of 65 rhizobial strains were successfully isolated from lentil plants grown in experimental fields in Jinzhong City, Shanxi Province. The colony growth phenotypes of these isolates were consistent with the basic characteristics of rhizobia. PCR amplification of the *nifH* nitrogen fixation gene indicated that these lentil rhizobial strains possess potential nitrogen-fixing capabilities. Following 16S rDNA sequencing and phylogenetic tree construction, the strains were found to belong to *Rhizobium* sp. and *Rhizobium leguminosarum.*

Most strains produced catalase, decomposing hydrogen peroxide into water and molecular oxygen, and exhibited acid and alkali production capabilities with BTB. In antibiotic resistance tests, the strains grew under low concentrations of ampicillin and streptomycin, but kanamycin significantly inhibited growth, with only seven strains resistant at 50 µg/mL. The majority of strains were more adapted to alkaline environments, and six strains could grow normally in 4% NaCl medium. However, the strains were not resistant to high or low temperatures. Two drought-resistant strains, TG 25 and TG 55, were identified in a PEG6000 simulated drought experiment. Inoculation with the dominant strain significantly increased the biomass of lentil plants in subsequent experiments. Therefore, this study will provide an important basis for the subsequent analysis of the mechanism of nitrogen fixation by lentil nodules, the enhancement of the efficiency of nitrogen fixation by nodules in lentil, and the development and utilization of rhizobial mycorrhizal agents in lentil.

## Figures and Tables

**Figure 1 microorganisms-13-01242-f001:**
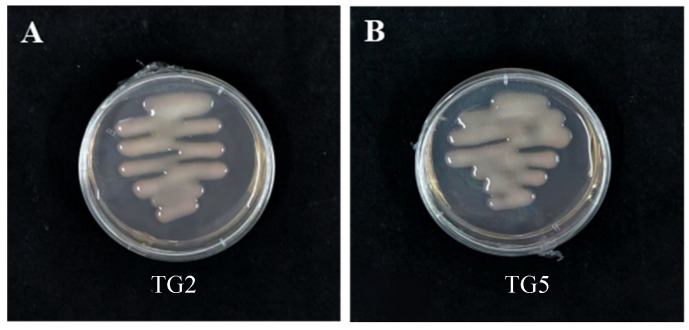
Isolation of rhizobium from lentil: (**A**,**B**) Growth of *Rhizobium leguminosarum* colonies in lentil TG2-TG5 strains.

**Figure 2 microorganisms-13-01242-f002:**
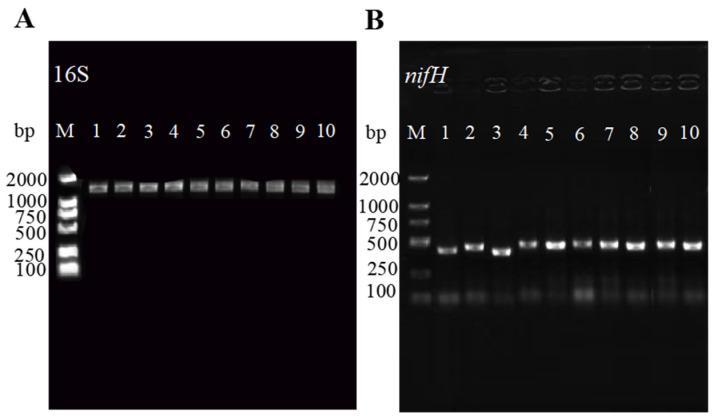
PCR amplification results of Rhizobium 16S rDNA and nitrogen fixation gene *nifH*: (**A**) 16S rDNA amplification results. (**B**) *nifH* nitrogen fixation gene amplification results. M: DL2000; 1: TG1; 2: TG2; 3: TG3; 4: TG4; 5: TG5; 6: TG6; 7: TG7; 8: TG8; 9: TG9; 10: TG10.

**Figure 3 microorganisms-13-01242-f003:**
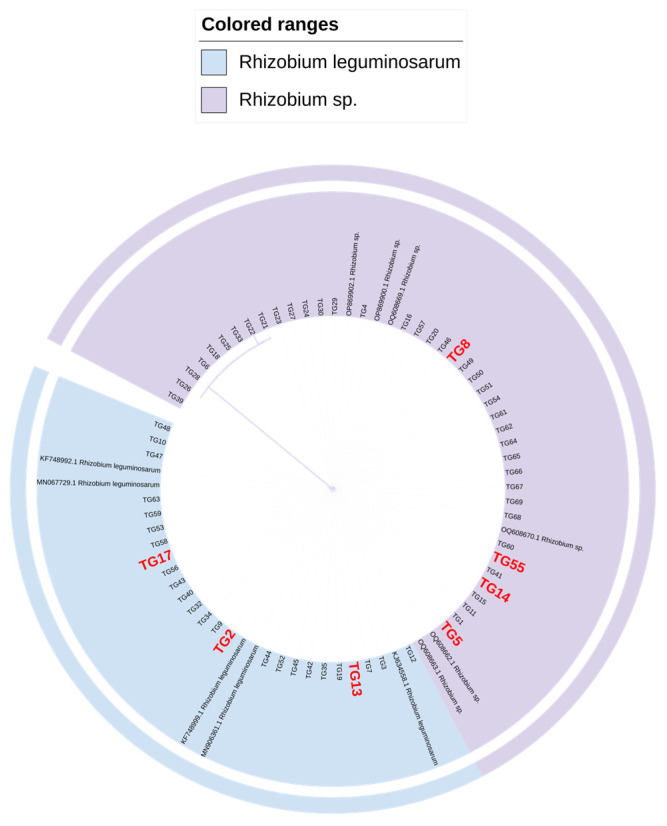
NJ phylogenetic tree of 16S rDNA sequences of Rhizobium.

**Figure 4 microorganisms-13-01242-f004:**
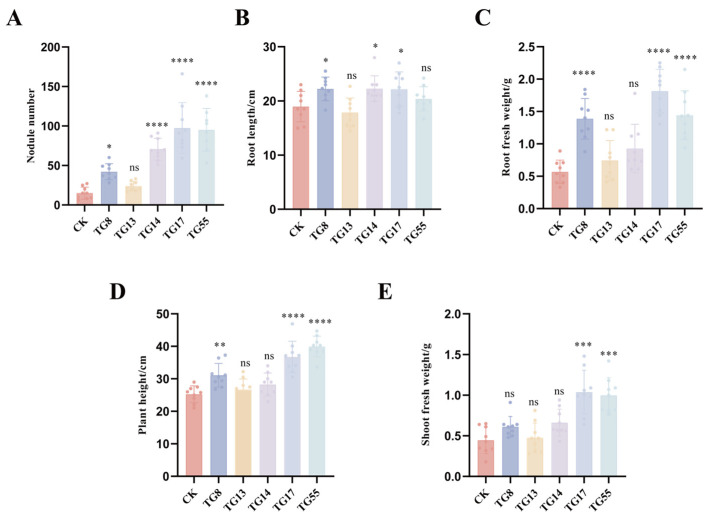
Phenotypic and growth index measurements of Rhizobium re-inoculation growth status: (**A**–**E**) are growth indexes measured after Rhizobium inoculation: (**A**) is the number of nodules (nos.); (**B**) is the root length (cm); (**C**) is the fresh weight of roots (g); (**D**) is the plant height (cm); and (**E**) is the fresh weight of shoots (g). Student’s *t*-tests were performed (ns: no significant difference. * *p* < 0.05, ** *p* < 0.01, and *** *p* < 0.001, **** *p* < 0.001, n = 6).

**Table 1 microorganisms-13-01242-t001:** Primers used for the identification of rhizobium genes.

Primer Name	Primer Sequence Information (5′–3′)	Reference
*nifH*-F	AAAGGYGGWATCGGYAARTCCACCAC	[17]
*nifH*-R	TTGTTSGCSGCRTACATSGCCATCAT	
16S rDNA-F	AGAGTTTGATCCTGGCTCAG	[18]
16S rDNA-R	GGTTACCTTGTTACGACTT	

**Table 2 microorganisms-13-01242-t002:** Results of the physiological and biochemical reactions of the strains.

Bacterial Strain	BTB Acid Production	BTB Base Production	Catalase Reaction	Bacterial Strain	BTB Acid Production	BTB Base Production	Catalase Reaction
*TG1*	+		+	*TG35*	+		+
*TG2*	+		+	*TG39*	-	-	+
*TG3*	+		-	*TG40*	+		+
*TG4*	+		+	*TG41*		+	+
*TG5*	+		+	*TG42*	+		+
*TG6*	+		-	*TG43*	+		+
*TG7*	+		+	*TG44*		+	+
*TG8*	+		-	*TG45*	+		+
*TG9*	+		+	*TG46*	+		+
*TG10*	+		-	*TG47*	+		+
*TG11*	+		+	*TG48*	+		+
*TG12*	+		+	*TG49*	+		+
*TG13*		+	+	*TG50*	+		+
*TG14*	+		+	*TG51*	+		+
*TG15*	+		+	*TG52*	+		+
*TG16*		+	+	*TG53*	+		+
*TG17*	+		+	*TG54*	+		+
*TG18*	+		+	*TG55*	+		+
*TG19*	+		+	*TG56*		+	+
*TG20*	+		+	*TG57*	+		+
*TG21*	+		+	*TG58*		+	+
*TG22*	+		+	*TG59*	+		+
*TG23*	+		+	*TG60*	+		-
*TG24*	+		+	*TG61*	+		+
*TG25*	+		+	*TG62*	+		+
*TG26*	+		+	*TG63*	+		+
*TG27*	+		+	*TG64*	+		+
*TG28*	+		+	*TG65*	+		+
*TG29*	+		+	*TG66*	+		-
*TG30*	+		+	*TG67*	+		-
*TG32*	+		-	*TG68*	+		+
*TG33*	+		+	*TG69*	+		+
*TG34*		+	+				

Notes: ‘+’ is a positive reaction and ‘-’ is a negative reaction.

**Table 3 microorganisms-13-01242-t003:** Results of strain antibiotic resistance identification (µg mL^−1^).

Bacterial Strain	Ampicillin			Streptomycin			Kanamycin		
	50	100	200	50	100	200	50	100	200
*TG1*	+	+	+	+	+	+	+	-	-
*TG2*	+	+	-	+	-	-	-	-	-
*TG3*	+	+	+	+	+	+	+	+	+
*TG4*	+	+	-	+	-	-	-	-	-
*TG5*	+	+	-	-	-	-	-	-	-
*TG6*	+	+	+	+	+	+	+	+	+
*TG7*	+	+	+	+	+	+	+	+	+
*TG8*	+	+	+	+	+	+	+	+	+
*TG9*	+	+	-	+	-	-	-	-	-
*TG10*	+	+	-	+	-	-	-	-	-
*TG11*	+	+	+	+	-	-	-	-	-
*TG12*	+	+	+	+	-	-	-	-	-
*TG13*	+	+	+	+	+	-	-	-	-
*TG14*	+	+	+	+	+	+	+	+	+
*TG15*	+	+	-	+	-	-	-	-	-
*TG16*	+	+	-	+	-	-	-	-	-
*TG17*	+	+	-	+	-	-	-	-	-
*TG18*	+	+	+	+	-	-	-	-	-
*TG19*	+	+	+	+	-	-	-	-	-
*TG20*	+	+	+	+	-	-	-	-	-
*TG21*	+	+	+	+	-	-	-	-	-
*TG22*	+	+	+	+	-	-	-	-	-
*TG23*	+	+	+	+	-	-	-	-	-
*TG24*	+	+	+	+	-	-	-	-	-
*TG25*	+	+	+	+	-	-	-	-	-
*TG26*	+	+	+	+	-	-	-	-	-
*TG27*	+	+	+	+	-	-	-	-	-
*TG28*	+	+	+	+	-	-	-	-	-
*TG29*	+	+	+	+	-	-	-	-	-
*TG30*	+	+	+	+	-	-	-	-	-
*TG32*	+	+	+	+	-	-	-	-	-
*TG33*	+	+	-	+	-	-	-	-	-
*TG34*	+	+	+	+	+	-	-	-	-
*TG35*	+	+	-	+	-	-	-	-	-
*TG39*	+	+	-	+	-	-	-	-	-
*TG40*	+	+	-	+	-	-	-	-	-
*TG41*	+	+	-	+	-	-	-	-	-
*TG42*	+	+	-	+	-	-	-	-	-
*TG43*	+	+	-	+	-	-	-	-	-
*TG44*	+	+	+	+	-	-	-	-	-
*TG45*	+	+	-	+	-	-	-	-	-
*TG46*	+	+	+	+	-	-	-	-	-
*TG47*	+	+	+	+	-	-	-	-	-
*TG48*	+	+	+	+	-	-	-	-	-
*TG49*	+	+	+	+	-	-	-	-	-
*TG50*	+	+	+	+	-	-	-	-	-
*TG51*	+	+	+	+	-	-	-	-	-
*TG52*	+	+	+	+	-	-	-	-	-
*TG53*	+	+	+	+	-	-	-	-	-
*TG54*	+	+	+	+	-	-	-	-	-
*TG55*	+	+	+	+	-	-	-	-	-
*TG56*	+	+	+	+	-	-	-	-	-
*TG57*	+	+	-	+	-	-	-	-	-
*TG58*	+	+	+	+	+	-	-	-	-
*TG59*	+	+	-	+	-	-	-	-	-
*TG60*	+	+	+	+	+	+	-	-	-
*TG61*	+	+	+	+	+	+	-	-	-
*TG62*	+	+	+	+	+	-	-	-	-
*TG63*	+	+	+	+	+	-	-	-	-
*TG64*	+	+	+	+	+	+	-	-	-
*TG65*	+	+	+	+	+	+	-	-	-
*TG66*	+	+	+	+	+	+	-	-	-
*TG67*	+	+	+	+	+	+	+	+	+
*TG68*	+	+	+	+	+	+	-	-	-
*TG69*	+	+	+	+	+	+	-	-	-

Notes: ‘+’ is a positive reaction and ‘-’ is a negative reaction.

**Table 4 microorganisms-13-01242-t004:** Screening of bacterial strains for salinity tolerance, acid/alkali tolerance, and temperature adaptability.

Bacterial Strain	Salt Tolerance					Tolerance to Acid and Alkali					Temperature Tolerance			
	0.50%	1%	1.50%	2%	4%	4	5	7	9	10	4 °C	28 °C	37 °C	49 °C
*TG1*	+	-	-	-	-	-	+	+	+	-	-	+	-	-
*TG2*	+	-	-	-	-	-	+	+	-	-	-	+	-	-
*TG3*	+	+	+	+	-	-	-	+	+	-	-	+	+	-
*TG4*	+	-	-	-	-	-	-	+	+	-	-	+	+	-
*TG5*	+	-	-	-	-	-	+	+	+	+	-	+	+	-
*TG6*	+	+	+	+	+	-	+	+	+	+	-	+	+	-
*TG7*	+	+	+	+	+	-	-	+	+	-	-	+	+	-
*TG8*	+	+	+	+	+	-	-	+	+	-	-	+	+	-
*TG9*	+	-	-	-	-	-	+	+	-	-	-	+	+	-
*TG10*	+	-	-	-	-	-	+	+	-	-	-	+	+	-
*TG11*	+	-	-	-	-	-	+	+	-	-	-	+	-	-
*TG12*	+	-	-	-	-	-	+	+	-	-	-	+	+	-
*TG13*	+	+	+	+	+	-	+	+	+	+	-	+	+	-
*TG14*	+	-	-	-	-	-	+	+	+	+	-	+	+	-
*TG15*	+	-	-	-	-	-	-	+	-	-	-	+	+	-
*TG16*	+	-	-	-	-	-	-	+	-	-	-	+	+	-
*TG17*	+	+	+	+	+	-	+	+	+	+	-	+	+	-
*TG18*	+	-	-	-	-	-	+	+	-	-	-	+	+	-
*TG19*	+	-	-	-	-	-	-	+	-	-	-	+	+	-
*TG20*	+	-	-	-	-	-	-	+	-	-	-	+	+	-
*TG21*	+	-	-	-	-	-	+	+	-	-	-	+	+	-
*TG22*	+	-	-	-	-	-	+	+	-	-	-	+	+	-
*TG23*	+	-	-	-	-	-	+	+	-	-	-	+	+	-
*TG24*	+	-	-	-	-	-	-	+	-	-	-	+	+	-
*TG25*	+	-	-	-	-	-	+	+	-	-	-	+	+	-
*TG26*	+	-	-	-	-	-	+	+	-	-	-	+	+	-
*TG27*	+	-	-	-	-	-	+	+	-	-	-	+	+	-
*TG28*	+	-	-	-	-	-	+	+	-	-	-	+	+	-
*TG29*	+	-	-	-	-	-	+	+	-	-	-	+	+	-
*TG30*	+	-	-	-	-	-	+	+	-	-	-	+	+	-
*TG32*	+	+	+	+	-	-	+	+	-	+	-	+	+	-
*TG33*	+	-	-	-	-	-	-	+	-	-	-	+	+	-
*TG34*	+	-	-	-	-	-	-	+	+	-	-	+	+	-
*TG35*	+	+	+	+	+	-	-	+	-	-	-	+	+	-
*TG39*	+	-	-	-	-	-	-	+	-	-	-	+	+	-
*TG40*	+	-	-	-	-	-	-	+	-	-	-	+	+	-
*TG41*	+	-	-	-	-	-	-	+	-	-	-	+	+	-
*TG42*	+	-	-	-	-	-	+	+	-	-	-	+	+	-
*TG43*	+	-	-	-	-	-	+	+	-	-	-	+	+	-
*TG44*	+	-	-	-	-	-	+	+	-	-	-	+	+	-
*TG45*	+	-	-	-	-	-	+	+	-	-	-	+	+	-
*TG46*	+	-	-	-	-	-	-	+	-	-	-	+	+	-
*TG47*	+	-	-	-	-	-	-	+	+	-	-	+	+	-
*TG48*	+	-	-	-	-	-	-	+	-	-	-	+	+	-
*TG49*	+	-	-	-	-	-	-	+	-	-	-	+	+	-
*TG50*	+	-	-	-	-	-	-	+	-	-	-	+	+	-
*TG51*	+	-	-	-	-	-	-	+	-	-	-	+	+	-
*TG52*	+	-	-	-	-	-	+	+	-	-	-	+	+	-
*TG53*	+	-	-	-	-	-	-	+	+	-	-	+	+	-
*TG54*	+	-	-	-	-	-	-	+	+	-	-	+	+	-
*TG55*	+	-	-	-	-	-	+	+	+	-	-	+	+	-
*TG56*	+	-	-	-	-	-	+	+	+	-	-	+	+	-
*TG57*	+	+	+	+	+	-	-	+	+	-	-	+	+	-
*TG58*	+	-	-	-	-	-	-	+	+	-	-	+	+	-
*TG59*	+	-	-	-	-	-	-	+	+	-	-	+	+	-
*TG60*	+	+	+	+	-	-	+	+	+	+	-	+	+	-
*TG61*	+	+	+	+	-	-	+	+	+	+	-	+	+	-
*TG62*	+	-	-	-	-	-	-	+	-	-	-	+	+	-
*TG63*	+	-	-	-	-	-	-	+	+	-	-	+	+	-
*TG64*	+	+	+	+	-	-	+	+	+	+	-	+	+	-
*TG65*	+	+	+	+	-	-	+	+	+	+	-	+	+	-
*TG66*	+	+	+	+	-	-	+	+	+	+	-	+	+	-
*TG67*	+	+	+	+	-	-	+	+	+	+	-	+	+	-
*TG68*	+	+	+	+	-	-	+	+	+	+	-	+	+	-
*TG69*	+	+	+	+	-	-	+	+	+	+	-	+	+	-

Notes: ‘+’ is a positive reaction and ‘-’ is a negative reaction.

**Table 5 microorganisms-13-01242-t005:** Growth of bacterial strains under PEG6000 stress conditions.

Bacterial Strain	CK(OD)	5%(OD)	10%(OD)	20%(OD)	30%(OD)
TG1	0.928 ± 0.034a	0.800 ± 0.048b	0.541 ± 0.017c	0.354 ± 0.101d	0.173 ± 0.072e
TG2	0.908 ± 0.127a	0.655 ± 0.078b	0.467 ± 0.095bc	0.371 ± 0.123c	0.150 ± 0.106d
TG3	0.928 ± 0.085a	0.727 ± 0.100b	0.464 ± 0.053bc	0.346 ± 0.096c	0.120 ± 0.014d
TG4	0.826 ± 0.110a	0.678 ± 0.090b	0.559 ± 0.119c	0.362 ± 0.109c	0.088 ± 0.104d
TG5	0.836 ± 0.105a	0.749 ± 0.097ab	0.509 ± 0.054b	0.346 ± 0.088c	0.117 ± 0.053d
TG6	0.836 ± 0.082a	0.732 ± 0.024a	0.505 ± 0.043b	0.346 ± 0.119c	0.110 ± 0.021d
TG7	0.85 ± 0.089a	0.728 ± 0.022a	0.517 ± 0.029b	0.379 ± 0.118c	0.109 ± 0.012d
TG8	0.87 ± 0.065a	0.717 ± 0.019b	0.519 ± 0.029c	0.386 ± 0.091d	0.109 ± 0.014e
TG9	0.87 ± 0.077a	0.726 ± 0.047b	0.526 ± 0.114c	0.342 ± 0.166d	0.098 ± 0.012e
TG10	0.921 ± 0.122a	0.724 ± 0.071a	0.431 ± 0.015b	0.314 ± 0.080c	0.110 ± 0.008d
TG11	0.854 ± 0.046a	0.722 ± 0.064b	0.535 ± 0.097c	0.364 ± 0.134c	0.112 ± 0.009d
TG12	0.881 ± 0.111a	0.547 ± 0.069a	0.437 ± 0.039b	0.328 ± 0.121c	0.145 ± 0.054d
TG13	0.959 ± 0.115a	0.784 ± 0.026b	0.435 ± 0.01bc	0.411 ± 0.076c	0.146 ± 0.036d
TG14	0.882 ± 0.102a	0.597 ± 0.045b	0.517 ± 0.083c	0.372 ± 0.125c	0.097 ± 0.089d
TG15	0.829 ± 0.079a	0.723 ± 0.051b	0.559 ± 0.058bc	0.339 ± 0.087c	0.098 ± 0.081d
TG16	0.97 ± 0.255a	0.635 ± 0.023a	0.572 ± 0.193b	0.373 ± 0.151c	0.073 ± 0.063d
TG17	0.771 ± 0.013a	0.592 ± 0.075b	0.423 ± 0.04b	0.294 ± 0.087b	0.063 ± 0.050c
TG18	0.859 ± 0.074a	0.651 ± 0.030b	0.481 ± 0.068c	0.353 ± 0.099d	0.100 ± 0.050e
TG19	0.847 ± 0.059a	0.771 ± 0.140b	0.481 ± 0.117c	0.366 ± 0.111d	0.105 ± 0.004e
TG20	0.860 ± 0.031a	0.697 ± 0.097a	0.495 ± 0.057b	0.391 ± 0.142b	0.115 ± 0.042c
TG21	0.836 ± 0.057a	0.718 ± 0.074b	0.502 ± 0.096c	0.459 ± 0.192c	0.096 ± 0.083d
TG22	0.837 ± 0.068a	0.611 ± 0.05a	0.572 ± 0.102b	0.368 ± 0.114b	0.114 ± 0.055c
TG23	0.815 ± 0.035a	0.757 ± 0.025b	0.412 ± 0.168b	0.339 ± 0.140c	0.122 ± 0.025d
TG24	0.823 ± 0.034a	0.752 ± 0.038a	0.513 ± 0.059b	0.339 ± 0.128b	0.134 ± 0.041c
TG25	0.794 ± 0.031a	0.684 ± 0.014a	0.547 ± 0.08b	0.365 ± 0.138c	0.110 ± 0.073d
TG26	0.81 ± 0.058a	0.684 ± 0.027ab	0.535 ± 0.073b	0.345 ± 0.109c	0.099 ± 0.032d
TG27	0.882 ± 0.106a	0.659 ± 0.035b	0.554 ± 0.010c	0.356 ± 0.089d	0.127 ± 0.040e
TG28	0.849 ± 0.055a	0.656 ± 0.052b	0.526 ± 0.062b	0.341 ± 0.128c	0.114 ± 0.061d
TG29	0.827 ± 0.014a	0.683 ± 0.039b	0.518 ± 0.277b	0.355 ± 0.117c	0.116 ± 0.041d
TG30	0.89 ± 0.064a	0.738 ± 0.06ab	0.591 ± 0.024bc	0.366 ± 0.181cd	0.108 ± 0.018d
TG32	0.893 ± 0.137a	0.785 ± 0.100ab	0.519 ± 0.034b	0.309 ± 0.110c	0.112 ± 0.056d
TG33	0.847 ± 0.060a	0.628 ± 0.042a	0.427 ± 0.116b	0.392 ± 0.110c	0.089 ± 0.046d
TG34	0.856 ± 0.067a	0.721 ± 0.124b	0.565 ± 0.062c	0.33 ± 0.151c	0.081 ± 0.073d
TG35	0.846 ± 0.069a	0.692 ± 0.061ab	0.574 ± 0.063b	0.359 ± 0.138c	0.102 ± 0.033d
TG39	0.803 ± 0.069a	0.628 ± 0.086b	0.567 ± 0.115b	0.38 ± 0.146c	0.121 ± 0.047d
TG40	0.913 ± 0.099a	0.687 ± 0.048ab	0.499 ± 0.11b	0.381 ± 0.121c	0.124 ± 0.042d
TG41	0.844 ± 0.046a	0.613 ± 0.033b	0.518 ± 0.034c	0.305 ± 0.091c	0.091 ± 0.032d
TG42	0.911 ± 0.027a	0.727 ± 0.063b	0.529 ± 0.108b	0.315 ± 0.113c	0.074 ± 0.033d
TG43	0.878 ± 0.029a	0.778 ± 0.073b	0.578 ± 0.081c	0.304 ± 0.134d	0.099 ± 0.022e
TG44	0.911 ± 0.087a	0.627 ± 0.076a	0.617 ± 0.07b	0.33 ± 0.098c	0.097 ± 0.029d
TG45	0.842 ± 0.041a	0.665 ± 0.034b	0.544 ± 0.062b	0.366 ± 0.151c	0.093 ± 0.032d
TG46	0.827 ± 0.086a	0.617 ± 0.127b	0.518 ± 0.118b	0.32 ± 0.134c	0.105 ± 0.010d
TG47	0.876 ± 0.095a	0.676 ± 0.022b	0.459 ± 0.026b	0.343 ± 0.115c	0.148 ± 0.067d
TG48	0.765 ± 0.068a	0.574 ± 0.037b	0.471 ± 0.105c	0.347 ± 0.11c	0.123 ± 0.016d
TG49	0.856 ± 0.097a	0.644 ± 0.05b	0.469 ± 0.002bc	0.333 ± 0.091c	0.099 ± 0.044d
TG50	0.798 ± 0.090a	0.506 ± 0.077b	0.502 ± 0.07c	0.321 ± 0.137d	0.153 ± 0.049e
TG51	0.807 ± 0.093a	0.646 ± 0.131b	0.52 ± 0.105b	0.371 ± 0.128c	0.088 ± 0.057d
TG52	0.831 ± 0.045a	0.739 ± 0.153ab	0.484 ± 0.051bc	0.374 ± 0.093c	0.112 ± 0.022d
TG53	0.878 ± 0.030a	0.647 ± 0.062a	0.498 ± 0.07b	0.382 ± 0.096b	0.104 ± 0.019c
TG54	0.991 ± 0.198a	0.591 ± 0.082b	0.384 ± 0.054c	0.371 ± 0.05c	0.158 ± 0.058d
TG55	0.834 ± 0.038a	0.576 ± 0.117b	0.562 ± 0.114c	0.37 ± 0.156d	0.146 ± 0.038e
TG56	0.805 ± 0.086a	0.575 ± 0.082b	0.486 ± 0.044b	0.334 ± 0.128c	0.101 ± 0.027d
TG57	0.916 ± 0.089a	0.664 ± 0.042b	0.612 ± 0.149b	0.353 ± 0.089c	0.121 ± 0.005d
TG58	0.841 ± 0.045a	0.656 ± 0.027b	0.459 ± 0.054b	0.345 ± 0.084c	0.123 ± 0.058d
TG59	0.907 ± 0.046a	0.631 ± 0.056b	0.499 ± 0.138c	0.397 ± 0.161d	0.111 ± 0.079e
TG60	0.956 ± 0.041a	0.691 ± 0.058b	0.572 ± 0.110bc	0.34 ± 0.137c	0.162 ± 0.018d
TG61	0.938 ± 0.082a	0.716 ± 0.099b	0.47 ± 0.044b	0.341 ± 0.133c	0.146 ± 0.017d
TG62	0.771 ± 0.049a	0.599 ± 0.083b	0.447 ± 0.044c	0.347 ± 0.154c	0.099 ± 0.045d
TG63	0.835 ± 0.055a	0.664 ± 0.117b	0.357 ± 0.127bc	0.331 ± 0.123c	0.099 ± 0.066d
TG64	1.002 ± 0.042a	0.737 ± 0.116a	0.452 ± 0.038b	0.269 ± 0.077b	0.145 ± 0.020c
TG65	1.079 ± 0.116a	0.637 ± 0.015b	0.431 ± 0.059c	0.334 ± 0.125d	0.142 ± 0.045e
TG66	0.956 ± 0.02a	0.629 ± 0.061b	0.489 ± 0.062c	0.335 ± 0.104c	0.139 ± 0.041d
TG67	0.969 ± 0.112a	0.673 ± 0.044b	0.447 ± 0.018c	0.343 ± 0.133d	0.122 ± 0.033e
TG68	1.049 ± 0.192a	0.708 ± 0.141b	0.461 ± 0.042c	0.309 ± 0.097cd	0.131 ± 0.033d
TG69	0.939 ± 0.037a	0.641 ± 0.022b	0.489 ± 0.145c	0.315 ± 0.141c	0.131 ± 0.032d

Note: letters are the mean and standard deviation of the three replicates and the significant differences.

## Data Availability

The original contributions presented in this study are included in the article. Further inquiries can be directed to the corresponding authors.

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
