# Peer review of "Screening and Validation of Rhizobial Strains for Improved Lentil Growth"

_microorganisms, 2025, doi:10.3390/microorganisms13061242_

Round 1

Reviewer 1 Report

Comments and Suggestions for Authors

Dear Authors,

In this paper, the authors identify a series of bacteria in the roots of lentils and study their resistance characteristics and their ability to improve certain physiological parameters in lentils. I believe the idea is good, but the way the data has been collected and presented is not appropriate. There are significant deficiencies that must be addressed. 

Majors:

-L77: Soil samples. Include the specific Cartesian coordinates of the location.

-The CR-YMA method is not highly specific. Discuss its limitations and disadvantages, please.

-L137: …..Reaction. A reference is needed.

-L143: …..Result. A reference is needed

-L168: For this method,  references are needed.

 -In the lentil inoculation experiments, provide more detailed statistical data (e.g., p-values) to support claims of significant increases in biomass.

-In stress tolerance tests (salinity, pH, temperature), detail how bacterial growth was evaluated (e.g., was only OD600 measured, or were qualitative analyses also performed?).

-Specify if the experiments were performed with biological and technical replicates.

-L187: “subsp”??

-The way the results are introduced is not correct; a prior introduction is needed, such as mentioning what 65 isolates? Where do they come from?

-In the results, there is no reference to Fig. 1 nor an explanation of what TG2 and TG5 are.

-What does Fig. 1 contribute to the paper? What culture medium was used? Why are these two shown and not others?

-L199: Sorry, I don’t understand, and I don’t see the authors explain it. If 65 isolates were obtained, in the figure 2 and 3 why does it drop to 10? Based on what? Please explain.

-The figure legend for Figure 3 is incomplete; it is necessary to indicate the number of replicates and the bootstrap value.

-L219: “yielded negative results" respect to what?? Catalase I guess.

-L226: “3.4. Antibiotic resistance” What is the relevance of this data to the objective of the paper? I'm sorry, but I don't understand. How does it contribute to the objective? What is the importance of these strains being more or less resistant to certain antibiotics? In my opinion, the authors should properly justify this or remove the data.

-TG14 and TG58 – how is it possible that it is resistant to 50 and 200 µg/mL of streptomycin but not to 100? Please explain.

-L237: “Table 3,” or 4

-L238: “and the same strains also grew well at concentrations of 1% to 2% NaCl” I don't think this is the case, based on their data.

-L236:  Before presenting the data directly, the authors should briefly explain the rationale behind taking those measurements. What do these data contribute? They need to clearly explain how they support the objective of the study.

-In Table 5, what type of units are being used? I assume OD, but this should be clearly indicated in the table legend. A critical issue is that no statistical error (e.g., standard deviation) is reported, nor is the number of times the experiments were repeated. In my opinion, this needs to be addressed. In my opinion, the data in Table 5 should have been presented as line graphs, with each point showing its corresponding error bar.

Section 3.7-In my understanding, the key experiment that the authors have not conducted is quantifying the nitrogen-fixing capacity of each of the isolated strains, using the Acetylene Reduction Assay or another similar method.

L261: “TG 25 and TG 55”??? But why do they choose these two, if there are others that show a lower percentage of growth reduction? Also, in the summary, it mentions TG17—shouldn't this be corrected? This type of mistake is very serious

L270: “chosen as dominant”?? I'm sorry, but I don't understand this concept. Could you please explain it in more detail?

-Figure 4 does not specify how the statistical analysis of the data was performed or the P-value obtained.

-L270: “TG 13, TG 14, TG 17, and TG 55” -In Fig. 3, there is the phylogenetic tree for TG8, but what about the others—TG13, TG14, TG17, and TG55? In my understanding, it is essential for the authors to also include these four. They should be identified at the genus level and, if possible, at the species level as well.

-In my understanding, the discussion, with only 15 references, is very limited and incomplete. It should be expanded. I suggest the following approach to do so

-Discuss more broadly the biological implications of identifying strains tolerant to multiple stress factors. How applicable are these strains to different geographical regions or agricultural conditions?

-Recent interactions between Microalgal and Nitrogen-Fixing Bacterial Consortia in the soil have also been reviewed. How would this influence the rhizobia? Delve deeper into the biotechnological potential of these interactions Please discuss.

-Expand the discussion by comparing your results with similar studies in other legumes or geographical region.

Minors:

-L34: “crops[1].” Typo

-L197: “ (A)”Typo, dot missing.

Author Response

Reviewer 1:

Comment 1: L77: Soil samples. Include the specific Cartesian coordinates of the location.

Our response: Thank you for this valuable comment. You can see in the revised Materials and Methods, we added “Soil samples were collected from two locations in Taigu District, Jinzhong City, Shanxi Province: the lentil grown experimental fields in Agricultural crops station (37.42157°N,112.57109°E) and in Shenfeng Village, (37.43275°N,112.60202°E) of Taigu District. Six points of soil samples in each fields were collected at a 20cm depth of surface soil.

Comment 2: The CR-YMA method is not highly specific. Discuss its limitations and disadvantages, please.

Our response: Thank you. As you concerned, CR-YMA is not highly specific, different microorganisms have growth requirements, CR-YMA method reduces the reliability of the isolation process. The YMA medium supports the growth of a wide range of microorganisms, which increases the likelihood of contamination during the isolation process. In some cases, certain microorganisms may exhibit similar colony morphologies on YMA medium, making it difficult to distinguish them without microscopic examination. We added the limitations of YMA medium in discussion section.

Comment 3: -L137: …..Reaction. A reference is needed.

-L143: …..Result. A reference is needed

-L168: For this method,  references are needed.

Our response: Thank you, we added references for this description in the revised version.

RAFIQUE M, ALI A, NAVEED M, et al. Deciphering the Potential Role of Symbiotic Plant Microbiome and Amino Acid Application on Growth Performance of Chickpea Under Field Conditions [J]. Front Plant Sci, 2022, 13: 852851.

OKYERE S K, WEN J, CUI Y, et al. Bacillus toyonensis SAU-19 and SAU-20 Isolated From Ageratina adenophora Alleviates the Intestinal Structure and Integrity Damage Associated With Gut Dysbiosis in Mice Fed High Fat Diet [J]. Front Microbiol, 2022, 13: 820236.

X.F. PEI. Screening of Drought-Tolerance Rhizobia of Soybean and Its proteomics [D], baoding, Hebei Agricultural University 2012.。

Comment 4: In stress tolerance tests (salinity, pH, temperature), detail how bacterial growth was evaluated (e.g., was only OD600 measured, or were qualitative analyses also performed?).

Our response: Thank you. For the osmotic stress tolerance of the strains was evaluated using different concentrations of PEG6000 (polyethylene glycol 6000).

Comment 5: Specify if the experiments were performed with biological and technical replicates.

Our response: Thank you. All the experiments performed three independent biological replicates,  we added this in figure or table legends.

Comment 6: S L187: “subsp”??

Our response: Thank you for point out this mistake. We have revised this into “Isolation and purification of Rhizobium spp.” in the revised version.

Comment 7: The way the results are introduced is not correct; a prior introduction is needed, such as mentioning what 65 isolates? Where do they come from?

Our response: Thank you. According to your suggestion, we added “Sixty-five Rhizobium strains were successfully isolated from lentil plants grown in soil samples from the Agricultural Crops Station and in Shenfeng Village of Taigu District,”

Comment 8: In the results, there is no reference to Fig. 1 nor an explanation of what TG2 and TG5 are.

Our response: Thank you for point out this, we cited the Fig 1 in the revised manuscript, added the explanations for TG2 and TG5 in the legend of Figure 1. “Isolation of rhizobium from lentil (A-B) Growth of Rhizobium leguminosarum colonies in lentil TG2-TG5 strains.”

Comment 9: What does Fig. 1 contribute to the paper? What culture medium was used? Why are these two shown and not others?

Our response: Thank you. Fig. 1 primarily explains the representative 65 isolated strains belong to the genera Rhizobium and Bradyrhizobium. The strains were cultured using YMA medium, and the NJ phylogenetic tree results in Figure 2 clearly show that TG2 and TG5 cluster together with the genera Rhizobium and Bradyrhizobium. Therefore, these two strains were selected for further study.

Comment 10: L199: Sorry, I don’t understand, and I don’t see the authors explain it. If 65 isolates were obtained, in the figure 2 and 3 why does it drop to 10? Based on what? Please explain.

Our response: Thank you.Based on the observed phenotypic characteristics of the strains, they were initially identified as Rhizobium. The 16s sequences of all 65 strains have been constructed into an evolutionary tree in Figure 3.

Comment 11: L219: “yielded negative result” respect to what?? Catalase I guess.

Our response: “yielded negative result”, It refers to the strains not producing bubbles under the catalase reaction with hydrogen peroxide, indicating a negative reaction.

Comment 12: “3.4. Antibiotic resistance” What is the relevance of this data to the objective of the paper? I'm sorry, but I don't understand. How does it contribute to the objective? What is the importance of these strains being more or less resistant to certain antibiotics? In my opinion, the authors should properly justify this or remove the data.

Our response: Thank you for this important comment. In nature, many bacteria exhibit varying degrees of resistance to antibiotic substances. This natural resistance is an external manifestation of bacterial genotypes and is related to the survival and competitive capabilities of bacteria in the environment. Antibiotic resistance has certain taxonomic value in rhizobia and is associated with the competitive ability of strains in nodule formation, which is why it is often used to analyze the diversity of rhizobial populations and for strain identification and classification. Additionally, most rhizobia are slow-growing strains that are easily contaminated, making the provision of antibiotic resistance essential for rhizobial cultivation and subsequent genetic manipulation of the strains.

Comment 13: TG14 and TG58 – how is it possible that it is resistant to 50 and 200 µg/mL of streptomycin but not to 100? Please explain.

Our response: Thank you for point out this mistake. We have corrected this in the revised version.

Comment 13: and the same strains also grew well at concentrations of 1% to 2% NaCl” I don't think this is the case, based on their data.

Our response: Thank you. We have repeated this experiments three times, the result tells that the same strains are tolerant to NaCl at concentrations of 1% and 2%.

Comment 14: -L236:  Before presenting the data directly, the authors should briefly explain the rationale behind taking those measurements. What do these data contribute? They need to clearly explain how they support the objective of the study.

Our response: Thank you for this comments. In the revised manuscript, we added “Rhizobium resistance contributes to strain isolation and purification and subsequent genetic manipulation.”

Comment 15: In Table 5, what type of units are being used? I assume OD, but this should be clearly indicated in the table legend. A critical issue is that no statistical error (e.g., standard deviation) is reported, nor is the number of times the experiments were repeated.

Our response: Thank you. In the revised manuscript, we added the clarification of units are OD. All this measurement repeated three times, we added the standard deviation for this data.

Comment 16:  “TG 25 and TG 55”??? But why do they choose these two, if there are others that show a lower percentage of growth reduction? Also, in the summary, it mentions TG17—shouldn't this be corrected? This type of mistake is very serious

Our response: Thank you. The selection of TG25 and TG55 as drought-resistant strains is based on their performance among the top 10 strains ranked by the reduction in growth after different PEG6000 treatments applied to all 65 strains; both TG25 and TG55 showed good performance. Therefore, these two strains were chosen as drought-resistant candidates. Additionally, we sincerely apologize for the incorrect in the abstract regarding the mention of TG17, which is a clear error, and we have corrected it in the revised manuscript.

Comment 17: “chosen as dominant”?? I'm sorry, but I don't understand this concept. Could you please explain it in more detail?

Our response: Thank you for the comment. In the revised manuscript, we modified this into “chosen as candidate strain”.

Comment 18: Figure 4 does not specify how the statistical analysis of the data was performed or the P-value obtained.

Our response: Thank you for this important comment. In the revised version, we specified the data collection and statistic analysis. We added“Phenotypic and growth index measurements of Rhizobium re-inoculation growth status A, B, C, D and E are growth indexes measured after Rhizobium inoculation, (A) is the number of nodules (nos. ); (B) is the root length (cm); (C) is the fresh weight of roots (g); (D) is the plant height (cm); and E is the fresh weight of shoots (g) Student's t-tests were performed (ns: no significant difference.,*p<0.05, **p<0.01, and ***p<0.001.****p < 0.001, n=6).****p < 0.001, n=6). ” in the figure legend.

Comment 19: L270: “TG 13, TG 14, TG 17, and TG 55” -In Fig. 3, there is the phylogenetic tree for TG8, but what about the others—TG13, TG14, TG17, and TG55? In my understanding, it is essential for the authors to also include these four. They should be identified at the genus level and, if possible, at the species level as well.

Our response: Thank you. This is a very important suggestion. The phylogenetic tree in Figure 3 was constructed using 10 randomly selected strains before conducting physiological and biochemical analyses to identify whether the isolated strains belong to the genus Rhizobium. Strains TG8, TG13, TG14, TG17, and TG55 were identified as dominant strains through physiological and biochemical analyses after confirming they belong to Rhizobium. Therefore, a phylogenetic tree was not constructed for these strains.

Comment 20: In my understanding, the discussion, with only 15 references, is very limited and incomplete. It should be expanded

Our response: Thank you. According to your suggestion, we have added a discussion in the revised manuscript regarding the biotechnological potential of further studying these interactions, as well as the biological significance of identifying strains that are tolerant to various stress factors. Relevant references have also been included.

WANG Z, SONG Y. Toward understanding the genetic bases underlying plant-mediated "cry for help" to the microbiota [J]. Imeta, 2022, 1(1): e8.

ATICI Ö, AYDıN İ, KARAKUS S, et al. Inoculating maize (Zea mays L.) seeds with halotolerant rhizobacteria from wild halophytes improves physiological and biochemical responses of seedlings to salt stress [J]. Biol Futur, 2025.

JI C, WANG X, H., LIU X, L. Research Progress on the Action Mechanism of Plant Growth-promoting Bacteria Under Salt Stress [J]. Biotechnology Bulletin,, 2020, 36(04): 131-43.

HAN Z, R., HUO Y, X., GUO S, Y. Mechanism and Industrial Application of Bacillus Tolerance to Stress Conditions [J]. Biotechnology Bulletin, 2024, 40(08): 24-38.

JING J, CONG W F, BEZEMER T M. Legacies at work: plant-soil-microbiome interactions underpinning agricultural sustainability [J]. Trends Plant Sci, 2022, 27(8): 781-92.

Reviewer 2 Report

Comments and Suggestions for Authors

Dear author, your manuscript presents an interesting and relevant biotechnological approach, focusing especially on the current need to search for sustainable biofertilizers and providing solutions in abiotic stress contexts. However, before considering publication, I recommend that you address some comments that could further improve the quality of your manuscript.

  1. Scientific names should be written in italics (genus and species), make this correction in line 32 “Lens culinaris MediK”
  2. In section 2.1 of materials and methods, I recommend briefly describing the process of taking soil samples, for example, at what depth and how many samples were taken.
  3. In line 96, 98 and, 172, change the term “sterile distilled water” to “sterilized distilled water”
  4. In section 2.3.1 in line 103 and in section 2.3.2, describe the process of long-term preservation of bacterial strains, which makes it look repetitive, I suggest keeping only one of the sections.
  5. In line 140 make use of subscripts in the hydrogen peroxide formula
  6. In section 2.3.7, indicate that PEG6000 is polyethylene glycol 6000
  7. In section 2.4, specify what type of statistical analysis was performed for the data treatment.
  8. Figure 1 is not cited in the text in section 3.1
  9. In Figure 1, the figure legend is redundant and repetitive.
  10. In line 237, change Table 3 to table 4.
  11. In section 3.6, table 5 is not mentioned or cited
  12. In figure 4, indicate in parentheses the unit of measurement in the label of the “y” axes of the graphs, e.g., nodule number (units), root length (cm), root fresh weight (g), etc.
  13. In figure 4, when including error bars, specify whether it is the standard deviation or the standard error. In the case of significant differences, indicate which post hoc test was performed.
  14. In the legend of Figure 4, it is not indicated how many replicates were made nor is it indicated if a statistical analysis such as an Anova was applied, remember that statistical analysis is the key to validate your results.
  15. I recommend restructuring the conclusion, since it seems like a general summary of your work.

I recommend emphasizing more on Why is this study important. And how does it relate to agricultural sustainability and food security, as well as whether the objectives stated in your introduction were achieved.

  1. With respect to the conclusion of your work, a good closing should recognize areas of improvement that you were able to detect throughout your study, or if there are future projections that would emphasize the applicability of your study.
  2. Please review the format of the reference list, try to use a homogeneous and adequate format since in some references the volume of the journal is omitted (reference 2), in some cases commas are missing to separate the different components of the reference.

In reference 14, the title of the article is incomplete and, in some cases, uses the name of the journal in its abbreviated form and in other cases uses the full name, consult the format proposed in the instructions for authors of MDPI.

Comments on the Quality of English Language

Overall, your manuscript is well written, but there are some phrases that sound forced or translated literally, For example:

“Lentils are rich in...” → you could use a more formal turn of phrase such as “Lentils are known for their high content of...”

“Back identification” → better to use “Reinoculation assay” or “Verification of inoculation effect”.

Author Response

Reviewer 2:

Comment 1: Scientific names should be written in italics (genus and species), make this correction in line 32 “Lens culinaris MediK”

Our response: Thank you. In the revised version, we corrected this into“Lens culinaris MediK”.

Comment 2In section 2.1 of materials and methods, I recommend briefly describing the process of taking soil samples, for example, at what depth and how many samples were taken.

Our response: Thank you. You can see in the revised Materials and Methods, we added “Soil samples were collected from two locations in Taigu District, Jinzhong City, Shanxi Province: the lentil grown experimental fields in Agricultural crops station (37.42157°N,112.57109°E) and in Shenfeng Village, (37.43275°N,112.60202°E) of Taigu District. Six points of soil samples in each fields were collected at a 20cm depth of surface soil.”

Comment 3In line 96, 98 and, 172, change the term “sterile distilled water” to “sterilized distilled water”

Our response:Thank you. In the revised version, we modified this into “sterilized distilled water”.

Comment 4In section 2.3.1 in line 103 and in section 2.3.2, describe the process of long-term preservation of bacterial strains, which makes it look repetitive, I suggest keeping only one of the sections.

Our response: Thank you for this important comments, according to your suggestion, we have removed the description of process of long-term preservation of bacterial strains in the revised manuscript, Section 2.3.2.

Comment 5In line 140 make use of subscripts in the hydrogen peroxide formula

Our response: Thank you. Revised into H2O2.

Comment 6In section 2.3.7, indicate that PEG6000 is polyethylene glycol 6000

Our response: Thank you. We revised this sentence into “The osmotic stress tolerance of the strains was evaluated using different concentrations of PEG6000 (polyethylene glycol 6000).”

Comment 7In section 2.4, specify what type of statistical analysis was performed for the data treatment.

Our response: Thank you. In the revised version, we added the Statistical analyses into the Materials and Methods section. Statistical analyses were performed using the SPSS statistical programme (version 23) and Student's t-tests were performed on graphs using GraphPad Prism version 8.0.0.

Comment 8: Figure 1 is not cited in the text in section 3.1

Our response: Thank you for point out this, we cited the Fig 1 in the revised manuscript.

Comment 9: In Figure 1, the figure legend is redundant and repetitive.

Our response: Thank you. We revised the legend of Fig.1 into “Isolation of rhizobium from lentil (A-B) Growth of Rhizobium leguminosarum colonies in lentil TG2-TG5 strains.”

Comment 10: In line 237, change Table 3 to table 4.

Our response: Thank you for point out this mistake. We corrected this in the manuscript.

Comment 11: In section 3.6, table 5 is not mentioned or cited

Our response: Thank you, we cited Table 5 in revised version at the proper position.

Comment 12: In figure 4, indicate in parentheses the unit of measurement in the label of the “y” axes of the graphs, e.g., nodule number (units), root length (cm), root fresh weight (g), etc.

Our response: Thank you. Changes have been made in the revised version. “Phenotypic and growth index measurements of Rhizobium re-inoculation growth status A, B, C, D and E are growth indexes measured after Rhizobium inoculation, (A) is the number of nodules (nos.); (B) is the root length (cm); (C) is the fresh weight of roots (g); (D) is the plant height (cm); and E is the fresh weight of shoots (g) Student's t-tests were performed (ns: no significant difference.,*p<0.05, **p<0.01, and ***p<0.001.****p < 0.001, n=6).”

Comment 13: In figure 4, when including error bars, specify whether it is the standard deviation or the standard error. In the case of significant differences, indicate which post hoc test was performed.

Our response: Thank you. Values are standard deviations and a Student's t-test was performed (***p < 0.001, n=6).

Comment 14: In the legend of Figure 4, it is not indicated how many replicates were made nor is it indicated if a statistical analysis such as an Anova was applied, remember that statistical analysis is the key to validate your results.

Our response: Thank you very much for this valuable comment. In the revised legend, we clarified the three independent replicates, variance analysis were performed using GraphPad Prism version 8.0.0.

Comment 15: I recommend restructuring the conclusion, since it seems like a general summary of your work.

Our response: Thank you, we reorganized the conclusion section in the revised version.

Comment 16: Please review the format of the reference list, try to use a homogeneous and adequate format since in some references the volume of the journal is omitted (reference 2), in some cases commas are missing to separate the different components of the reference.

Our response: Thank you. According to your suggestion, we checked and unified the references of the manuscript and changed reference 2 into “X. C.SUN, J.TIAN, Z.W.ZHANG, et al.Application and Prospect of Biological Nitrogen Fixation on Agriculture. [J]. Hunan Agricultural Sciences,2019,5(11):16-20.”

Reviewer 3 Report

Comments and Suggestions for Authors

Dear Microorganisms

Editorial Office

I share my observation about the manuscript microorganisms-3602491 “Screening and Validation of Rhizobial Strains for Improved Lentil Productivity”. This manuscript showed a study about rhizobial strains of lentil. There is a lack of information in material and methods. The authors didn’t write about the depth of soil collection, the number of soil samples collected, whether they used a germination chamber, and the composition of the nutrient solution. The authors didn’t write where the plants grew and how they evaluated the root length, the root fresh weight, the plant height, and the shoot fresh weight. The statistical analysis wasn’t written adequately. Suggestions and doubts are in the manuscript.

Best regards!

Author Response

Reviewer 3:

Comment 1-2: What was the depth of soil collection?How many soil samples did you collect from each place? Didn't you collect in two locations (experimental field in Taigu District and the experimental field in 79 Shenfeng Village)?

Our response: Thank you for this crucial comments. You can see in the revised Materials and Methods, we added “Soil samples were collected from two locations in Taigu District, Jinzhong City, Shanxi Province: the lentil grown experimental fields in Agricultural crops station (37.42157°N,112.57109°E) and in Shenfeng Village, (37.43275°N,112.60202°E) of Taigu District. Six points of soil samples in each fields were collected at a 20cm depth of surface soil.”

Comment 3: How did you make the incubation? Was in a "germination chamber"?

Our response: Thank you. For the inoculation, Healthy seeds were selected, and the seeds were surface sterilized for 5 minutes using 75% ethanol and a 5% NaClO solution, followed by rinsing with sterile water 3 to 5 times. After treatment, the small lentil seeds were placed in petri dishes containing filter paper and incubated in a constant temperature incubator at 28℃.

Comment 4: What was the culture medium?

Our response: Thank you. The medium is a mixture of vermiculite and perlite in a volume ratio of 2:1.

Comment 5: What was the composition of the nutrient solution?

Our response: Thank you. We included the main components of the nutrient solution contained “0.03 g/L Ca(NO3)2·4H2O, 0.1 g/L CaCl2·2H2O, 0.1 g/L KH2PO4, 0.15 g/L Na2HPO4·12H2O, 0.12 g/L MgSO4·7H2O,        0.05 g/L Ferric Citrate, 2.86 mg/L H2BO3, 1.81 mg/L MnSO4, 0.22 mg/L ZnSO4, 0.8 mg/L CuSO4, 0.02 mg/L Na2MoO4·2H2O. ” in the revised version.

Comment 6: Did plants grow in the soil?

Our response: The plants are not grown in soil, but in a substrate made from a mixture of vermiculite and perlite.

Comment 7: Was experimental design in entirely random?

Our response: Experiments were designed entirely random.

Comment 8: What was the statistical analysis?Did you use a means test?

Our response: Thank you. statistical analyses were performed using the SPSS statistical programme (version 23) and graphs were subjected to Student's t-tests using GraphPad Prism version 8.0.0 to validate the differences in the manuscripts with respect to each growth indicator. *p<0.05, **p<0.01, ***p<0.001.

Comment 9: How did you make this assessment?

Our response: From Figure 4 (B), it can be observed that, compared to the uninoculated control plants (ck), the root lengths of TG8, TG14, and TG17 increased.

Comment 10: There was a field experiment? Where did the lentils grew?

Our response: Thank you. Field trials have not yet been conducted; the lentils were grown in pot experiments in a controlled temperature greenhouse.

Comment 11: Did you make a means test? Was it Tukey?

Our response: Thank you. The variance analysis and student's t-tests were used for multiple comparisons.

Round 2

Reviewer 1 Report

Comments and Suggestions for Authors

I believe the authors have adequately addressed all of my comments and suggestions, and I accept the paper in its current version.

Author Response

Thank you very much for your positive feedback for our manuscript. We are pleased to hear that you are satisfied with our response and revisions. Your approval means a lot to us. Your expertise and insights have been instrumental in improving the quality of our work.

Reviewer 3 Report

Comments and Suggestions for Authors

Dear Microorganisms

Editorial Office

I share my observation about the manuscript microorganisms-3602491 “Screening and Validation of Rhizobial Strains for Improved Lentil Productivity”. In this submission, the manuscript has new authors and the order of the authors was changed. Some points until aren’t clear. Is not there experimental design?  The statistical analysis is incomplete.

Best regards!

Author Response

We sincerely thank you for taking the time to review our manuscript amidst your busy schedule, and for providing valuable comments and suggestions.

We have attached great importance to your opinions and have developed a concrete revision to ensure further improvement in the quality of the paper.

During the revision process, we have carefully adjusted and improved the manuscript in accordance with your suggestions. With regard to the deficiencies you pointed out in statistical analysis and experimental design, we have made earnest supplements and optimizations .

Once again, we deeply appreciate the valuable comments you have provided on our manuscript. These insights are extremely precious to us and will greatly help improve the quality of our research and writing. We are confident that, through your thoughtful guidance and our dedicated revisions, the overall quality of the paper has been significantly enhanced.